# X-ray-diffraction and electrical-transport imaging of superconducting superhydride (La,Y)H$_{10}$

Abdul Haseeb Manayil Marathamkottil ●[1] ✉, Kui Wang[2], Nilesh P. Salke ●[2], Muhtar Ahart[2], Alexander C. Mark ●[2], Rostislav Hrubiak ●[3], Stella Chariton ●[4], Dean Smith ●[3], Vitali B. Prakapenka ●[4], Maddury Somayazulu ●[3], Nenad Velisavljevic ●[3,5] & Russell J. Hemley ●[1,2,6] ✉

Understanding how microscopic structural domains govern macroscopic electronic properties is central to advancing hydride superconductors, yet such correlations remain poorly resolved under pressure. We report the synthesis and characterization of (La$_{0.9}$Y$_{0.1}$)H$_{10}$ superhydrides exhibiting coexisting cubic $Fm\bar{3}m$ and hexagonal $P6_3/mmc$ clathrate phases observed over the pressure range from 168 GPa down to 136 GPa. Using synchrotron-based X-ray diffraction imaging at the upgraded Advanced Photon Source, we spatially resolved µm-scale distributions of these phases, revealing structural inhomogeneity across the sample. Four-probe resistance measurements confirmed superconductivity with two distinct transitions: an onset at 244 K associated with the cubic phase and a second near 220 K linked to the hexagonal phase. Notably, resistance profiles collected from multiple current and voltage permutations showed variations in transition width and onset temperature that correlated with the spatial phase distribution. These findings demonstrate a direct connection between local structural domains and superconducting behavior.

Dense hydrides exhibiting very high-temperature superconductivity are established as an extraordinary class of new materials of great fundamental and potential applied interest[1]. The remarkable quantum phenomena of these materials arise from the potential of dense hydrogen lattices to mimic atomic metallic hydrogen, which has long been predicted to exhibit room-temperature superconductivity under extreme pressures[2,3]. Advances in computational methods combined with experimental diamond anvil cell (DAC) techniques led to the discovery of rare-earth metal superhydrides with superconducting critical temperatures ($T_c$'s) approaching room temperature at megabar (>100 GPa) pressures[4–10]. In particular, lanthanum decahydride (LaH$_{10}$) was found to superconduct at temperatures up to 260 K at 188 GPa in the cubic $Fm\bar{3}m$ clathrate phase[8,9]. This structure features a three-dimensional hydrogen cage network surrounding La atoms, enabling the strong electron–phonon coupling origin of this very high $T_c$ superconductivity[4–7]. Notably, the near-room-temperature superconductivity was confirmed and has been independently reproduced by a growing number of groups[10–12].

Recent research has focused on strategies to enhance $T_c$ and stability of these materials by chemical substitution to form ternary and higher order phases[13–21]. Theoretical predictions for La superhydrides indicate that substituting a fraction of La with smaller elements such as yttrium (Y) chemically pre-compresses the lattice, modify phonon spectra, and extend the stability field of the clathrate

[1]Department of Chemistry, University of Illinois Chicago, Chicago, IL 60607, USA. [2]Department of Physics, University of Illinois Chicago, Chicago, IL 60607, USA. [3]HPCAT, X-ray Science Division, Argonne National Laboratory, Lemont, IL 60439, USA. [4]Center for Advanced Radiation Sources, University of Chicago, Chicago, IL 60637, USA. [5]Physics Division, Lawrence Livermore National Laboratory, Livermore, CA 94550, USA. [6]Department of Earth and Environmental Sciences, University of Illinois Chicago, Chicago, IL 60607, USA. ✉e-mail: amanay2@uic.edu; rhemley@uic.edu

phase[13–15]. In particular, (La,Y)H$_{10}$ phases are predicted to retain the high symmetry of LaH$_{10}$ while gaining enhanced structural stability through Y incorporation[13–16]. Experimental work has shown $T_c$ values up to 253 K for (La$_{0.8}$Y$_{0.2}$)H$_{10}$ at 183 GPa[22], yet a comprehensive understanding of the pressure–composition phase space remains incomplete[22,23].

Moreover, the coexistence of multiple structural phases together with stress-strain gradients and differences in synthesis conditions, in these and related samples further raises questions about how local structural heterogeneity affects superconducting behavior[6,9,10,21,22]. Traditional bulk measurements obscure this complexity by averaging over the entire sample volume, thereby masking domain-level inter- actions and local phase variations[24,25]. Resolving such structure–property relationships requires techniques that can spatially correlate structural and electronic properties, particularly under the extreme conditions in DAC experiments, where the length scales of structure–property variations fall in the μm to sub-μm range[26].

Synchrotron-based scanning X-ray diffraction microscopy (SXDM) has emerged as a powerful tool for probing structural heterogeneity in compressed materials[26–29]. Enabled by the upgraded Advanced Photon Source (APS-U), SXDM offers high spatial resolution and sensitivity to local symmetry variations, strain, and phase coexistence. Combined with X-ray diffraction imaging (XDI) for visualization, this approach provides a direct view of heterogeneous states that remain hidden in conventional diffraction measurements[26,29]. Complementing this microstructural information, four-probe direct-current (DC) resistance measurements with multiple current/voltage probe permutations enable spatial sampling of superconducting behavior across a sample[30]. Although originally designed for homogeneous sheets, the van der Pauw (VDP) method can be adapted to probe inhomogeneous materials and correlated with spatially resolved XDI measurements to capture broadened or multi-step transitions arising from structural inhomogeneity[30,31]. Such spatially resolved transport techniques have yet to be applied to hydrides owing to the challenges of DAC experiments.

In this work, we synthesized (La$_{0.9}$Y$_{0.1}$)H$_{10}$ at high pressure and employed SXDM in combination with multi-channel electrical trans- port measurements. Using a ~1 μm focused X-ray beam at HPCAT-U, SXDM revealed μm-scale structural inhomogeneity via XDI and cor- related it with local superconducting transitions captured via spatially resolved resistance measurements. This represents the first direct correlation between crystal structure and superconductivity in hydrides. Two distinct superconducting onsets near 244 K and 220 K were found to correspond to regions dominated by the cubic $Fm\bar{3}m$ and hexagonal $P6_3/mmc$ clathrate phases, respectively, enabling unambiguous assignment of $T_c$ values to specific structures. Yttrium substitution stabilizes the coexistence of both phases down to 136 GPa, without raising $T_c$ above that of pure LaH$_{10}$. Overall, our findings establish a clear, spatially resolved link between structural domains and superconducting behavior in hydrides.

## Results

### Structural characterization of (La,Y)H$_{10}$

Prior to laser heating, the La$_{0.9}$Y$_{0.1}$ alloy compressed together with ammonia borane (NH$_3$BH$_3$) to high pressure adopted a distorted-cubic $Fmmm$ structure[32]. At 158 GPa (DAC #1, Fig. 1), the X-ray diffrac- tion (XRD) pattern showed spotty diffraction rings, a narrower acces- sible 2θ range at the longer wavelength, and additional peaks from Pt electrodes and the sample environment, which obscured weaker La-Y reflections. By contrast, at 172 GPa (DAC #2, Fig. S1), the shorter wavelength, absence of electrode and gasket contributions, and improved powder averaging yielded well-resolved La-Y reflections. A direct comparison of La$_{0.9}$Y$_{0.1}$ at 158 and 172 GPa is provided in Fig. S2, showing that the observed differences in the number and shape of peaks arise from experimental factors rather than intrinsic structural

changes. In addition, the broader and asymmetric (002) profile at 172 GPa reflects pressure-induced lattice distortions and deviatoric stress, consistent with prior observations in elemental La under compression[32].

Following laser heating, the diffraction pattern changed sig- nificantly, with alloy peaks disappearing and new reflections emerging from hydrogen-rich phases (Figs. 1 and S1), confirming successful hydrogenation. At 158 GPa (Fig. 1), the pressure relaxed to 153 GPa after laser heating, and the diffraction pattern revealed the formation of two clathrate structures: cubic $Fm\bar{3}m$ and hexagonal $P6_3/mmc$, both of which have been previously reported in La–H and La–Y–H systems[10,22]. Optical images of the sample before and after laser heating are pro- vided in Fig. S3. A second synthesis performed at 172 GPa (Fig. S1) produced the same two phases, demonstrating reproducibility across independent runs.

Le Bail refinements of the hydrogenated sample at 153 GPa yielded lattice parameters of $a = 5.15(1)$ Å for the $Fm\bar{3}m$ phase and $a = 3.71(1)$ Å, $c = 5.54(1)$ Å for the $P6_3/mmc$ phase, corresponding to unit cell volumes of 136.6(1) Å$^3$ and 66.1(1) Å$^3$, respectively (Fig. 1, top panel). These values are comparable to those reported for undoped LaH$_{10}$ at similar pressures[6], with slight reductions in volume consistent with the expected lattice contraction from yttrium substitution[14,22]. As shown in Fig. S4, this systematic reduction provides direct structural evidence that ~10% Y is incorporated into the clathrate framework. Notably, no diffraction peaks from secondary LaH$_n$ or YH$_n$ phases were observed, indicating that Y substitution remains within the solubility limit for forming a single-phase clathrate or a mixed-phase clathrate solid solution[13,22]. Volume-based stoichiometry analysis yielded ~10 H per metal atom for both the cubic and hexagonal phases, consistent with the nominal (La$_{0.9}$Y$_{0.1}$)H$_{10}$ composition (Supplementary Note 1).

To assess the stability of the observed phases during decom- pression, additional XRD measurements were performed on both samples. In DAC #1, the sample was gradually decompressed from 153 GPa, and an XRD was collected at 136 GPa from the sample center, with Fig. S5 illustrating the collection positions corresponding to the patterns in Figs. 1 and S6. At 136 GPa, both the cubic and hexagonal phases were still observed (Figs. S6 and S7). Fig. S7 presents XRD patterns of (La,Y)H$_{10}$ at 153 and 136 GPa in DAC #1, collected from comparable positions in the sample chamber to confirm the persis- tence of both phases across this pressure range. At 136 GPa, minor distortions in the $Fm\bar{3}m$ phase were also evident across different regions of the sample (Fig. S8), likely reflecting local pressure gradients and lattice relaxation. This pressure lies near the known structural phase boundary of undoped LaH$_{10}$, where transitions to lower- symmetry structures such as $R\bar{3}m$ or $C2/m$ typically occur[6,12]. The persistence of both clathrate phases at 136 GPa suggests that Y sub- stitution extends the structural stability of LaH$_{10}$-type phases to lower pressures than observed in the undoped system[6,12].

In DAC #2, after initial synthesis at 172 GPa and characterization at 168 GPa, the sample was decompressed to 161 GPa. At this pressure, both the $Fm\bar{3}m$ and $P6_3/mmc$ phases are clearly present (Fig. S4), in contrast to binary La hydrides where the cubic phase typically dominates[6,10]. The coexistence of both clathrate phases at this stage of decompression underscores the role of Y substitution in stabilizing structural polymorphism beyond that observed in undoped LaH$_{10}$[6,9,10,12]. The repeated observation of both phases across the 168–136 GPa range demonstrates persistent coexistence rather than distinct, pressure-stabilized states. Their simultaneous presence indi- cates that ~10% Y substitution promotes polymorphic coexistence under our synthesis and decompression conditions, rather than favoring a single dominant structure. This contrasts with LaH$_{10}$, which transforms to lower-symmetry structures upon decompression[6,9–12]. The spatial variation and distribution of the cubic and hexagonal domains, further examined through diffraction imaging, highlight the complexity of phase coexistence near the clathrate stability boundary.

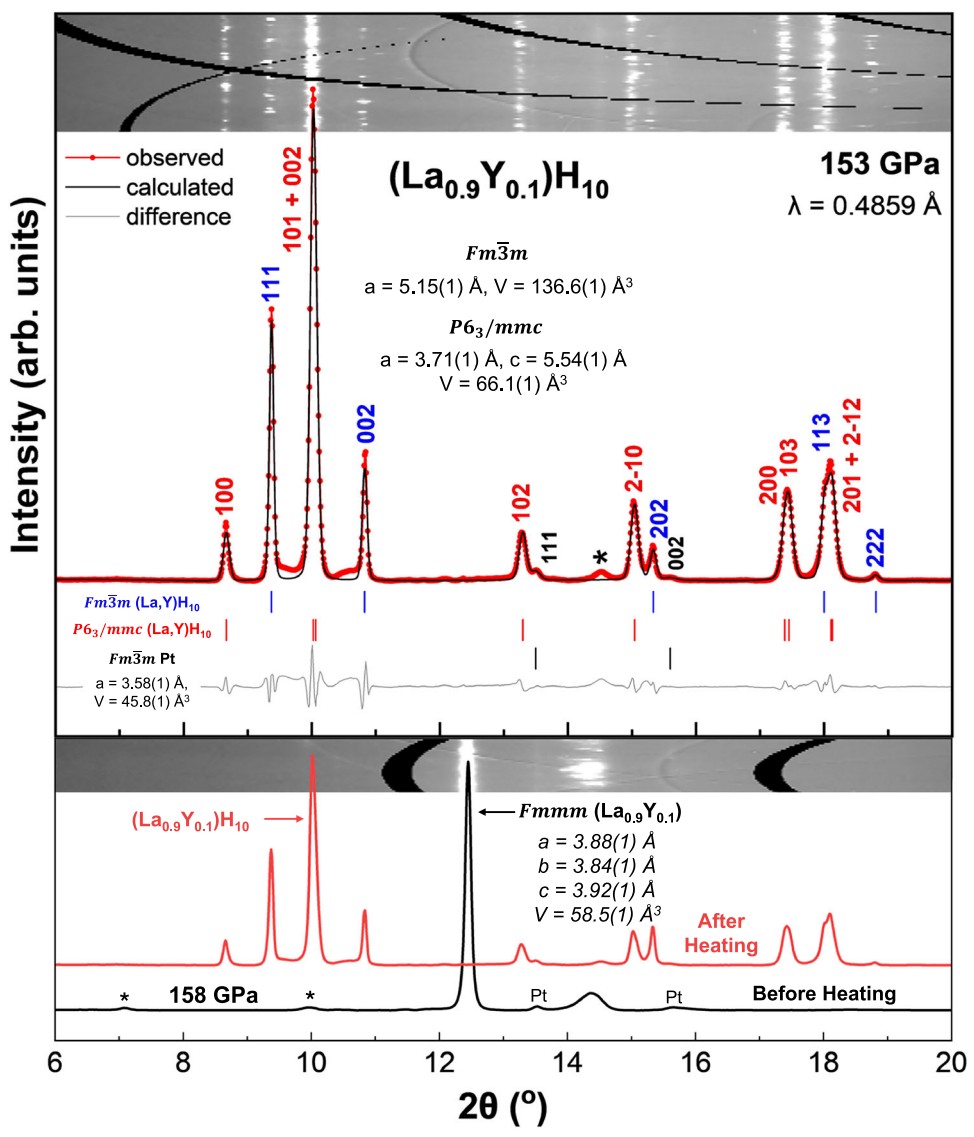

**Fig. 1 | XRD patterns and structural refinement of coexisting (La,Y)H₁₀ phases.**
Top: Experimental synchrotron XRD patterns and Le Bail refinements of the $Fm\bar{3}m$ and $P6_3/mmc$ (La,Y)H₁₀ phases at 153 GPa. The experimental data, fit, and residuals are shown in red, black, and gray, respectively. Refined lattice parameters for both phases are indicated. Bottom: Experimental XRD patterns of La₀.₉Y₀.₁ at 158 GPa before and after laser heating. The pre-heating pattern corresponds to the distorted-cubic $Fmmm$ phase. After laser heating, the diffraction pattern shows the formation of (La,Y)H₁₀ phases, consistent with the refined structures shown above. Reflections from Pt electrodes are explicitly labeled, while peaks marked with "*" correspond to unidentified or sample-environment contributions. Insets show representative 2D diffraction images for reference.

The pressure–volume ($P–V$) behavior of (La₀.₉Y₀.₁)H₁₀, shown in Fig. S4, generally follows the trend reported for undoped LaH₁₀, with modest phase-specific deviations that point to non-uniform compressibility and local strain effects. Similar trends were reported for (La₀.₈Y₀.₂)H₁₀ synthesized at higher pressures[22], where clathrate structures remained stable without decomposition. These results demonstrate that partial Y substitution preserves the hydrogen cage framework of LaH₁₀ while extending phase coexistence across a broader pressure range[22,23]. Such structural robustness provides the basis for linking local phase heterogeneity to superconducting behavior, as discussed in the following sections.

**Spatial mapping of structural domains via SXDM and XDI**
Spatial phase mapping at 153 GPa revealed μm-scale coexistence of $Fm\bar{3}m$ and $P6_3/mmc$ clathrate domains. Using SXDM at HPCAT-U, diffraction patterns were collected across a 30 × 30 μm² region with ~3 μm steps. XDI-based analysis identified phase-specific intensity distributions by integrating the first two Bragg reflections unique to each structure, producing two-dimensional maps of local phase domains. Figure 2 shows XDI maps at 153 GPa resolving the FCC and HCP phases, correlated with the 2D X-ray scan and the optical image of the sample after laser heating. Image analysis of the domain maps indicates that the cubic phase covers approximately 42% of the mapped region, while the hexagonal phase accounts for 58%.

The $Fm\bar{3}m$ phase is localized in discrete clusters near Pt leads #2, #3, and #4, with the largest fraction around lead #2 where excess ammonia borane was present (Fig. 2A). By correlating optical images before and after laser heating (Fig. S3) with the XDI maps, the interface between the sample and ammonia borane is identified between leads #2 and #3, consistent with regions of higher temperature and greater hydrogen availability favoring the $Fm\bar{3}m$ phase, while regions farther from this interface exhibit a higher fraction of the $P6_3/mmc$ phase, forming a continuous matrix between leads #1 and #4 (Fig. 2A). These correlations indicate that the observed structural inhomogeneity most

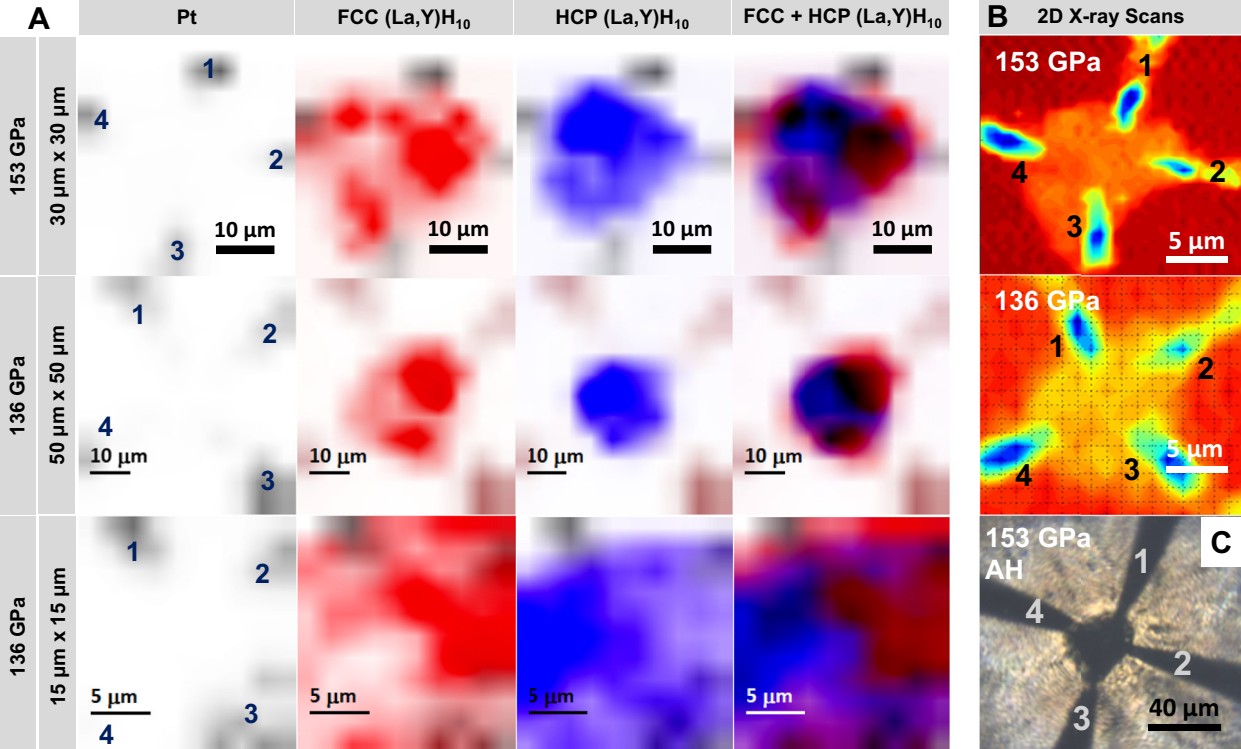

**Fig. 2 | Spatially resolved XRD imaging of phase domains in (La$_{0.9}$Y$_{0.1}$)H$_{10}$ at high pressure. A** XDI maps of the laser-heated (La,Y)H$_{10}$ sample at 153 GPa (top) and 136 GPa (middle and bottom), obtained via raster scanning with a ~1 μm focused synchrotron beam. The 30 × 30 μm² scan at 153 GPa resolves spatial distributions of cubic $Fm\bar{3}m$ (red) and hexagonal $P6_3/mmc$ (blue) domains, with Pt leads mapped in dark gray or brown. Electrode positions are annotated directly on the Pt to enable spatial correlation with electrical transport measurements. The rightmost column overlays all phases to show the composite spatial distribution. At 136 GPa, a 50 × 50 μm² scan (middle) and a higher-resolution 15 × 15 μm² scan (bottom) show continued coexistence of FCC and HCP domains with spatial variation. **B** 2D X-ray scan overview of the sample at respective pressures. **C** Optical image of the sample at 153 GPa after laser heating.

likely arises from variations in hydrogen content and laser heating geometry[9,10,12]. As shown in Fig. S9, the local pressure gradient does not correlate with FCC/HCP boundaries, indicating that pressure is not the dominant factor in domain arrangement at synthesis conditions. While phase coexistence is common in multiphase hydride systems[10,22], the ability to directly image μm-scale domain structure at this resolution provides a valuable framework for linking local structural environments with superconducting behavior, as discussed in the following sections.

Additional raster scans were performed at 136 GPa after decompression (Fig. 2A). Two maps, a broader 50 × 50 μm² grid and a focused 15 × 15 μm² grid, were collected from the same central region and are shown in Fig. 2. In the larger scan, the $Fm\bar{3}m$ phase exhibits reduced intensity near Pt lead #4 compared to the 153 GPa map, while the $P6_3/mmc$ phase remains more uniformly distributed across the sample chamber. The composite map again shows dominance of the hexagonal phase between leads #1 and #4, and clustering of the cubic phase around leads #2 and #3. The smaller scan offers a higher-resolution view of the local phase distribution and confirms the persistence of structural heterogeneity upon decompression.

The use of a micro-focused beam combined with SXDM enabled spatial mapping of phase-separated regions within the sample that may be challenging to resolve using conventional bulk XRD techniques. The spatial resolution in this study was chiefly governed by the ~1 μm beam size of APS-U and the small raster step size used during SXDM. Together, these parameters enabled fine spatial sampling across the sample chamber, allowing detection of μm-scale structural variations. Prior applications of XDI have demonstrated its effectiveness in visualizing structural gradients and preferred nucleation patterns in FeH$_x$[33], H$_3$S[34], H$_2$[35], and La–Y–Ce–H[36] systems. However, earlier

studies often had overlapping grids or had limited phase assignment capability due to reduced flux or detector sensitivity. The ability to resolve discrete $Fm\bar{3}m$ and $P6_3/mmc$ domains across the sample provides unique insight into structural heterogeneity. These spatially resolved maps form the foundation for linking local phase composition with superconducting behavior, as discussed below.

## Superconductivity in coexisting phases

Four-probe DC resistance measurements were carried out on the (La$_{0.9}$Y$_{0.1}$)H$_{10}$ sample in DAC #1 following structural characterization. As shown in Fig. 3A, resistance vs. temperature curves collected during warming cycles at four pressures between 153 and 136 GPa consistently display two distinct superconducting transitions. At 153 GPa, the first resistance drop begins at $T_{c,onset}$ = 244 K, followed by a second transition near $T_{c,onset}$ = 220 K (Fig. S10). The total transition width of $\Delta T \approx 28$ K is unusually broad for DC transport measurement and is characteristic of phase coexistence and electronic heterogeneity[37].

To further investigate the origin of these transitions, the temperature dependence of eight partial resistance traces [$R_{ab,cd}(T)$] were collected using the standard VDP permutations, involving four voltage pairs and two current paths, as shown in Fig. 4. Each measurement configuration was overlaid onto the composite XDI phase map, enabling direct spatial correlation between structural domains and electronic behavior. Notably, configurations such as $R_{34,12}$, $R_{34,21}$, $R_{41,23}$, and $R_{41,32}$ (Fig. 4B, D) exhibited a sharp superconducting drop near 240 K, with a narrow transition width of $\Delta T < 10$ K. These measurements probed regions between electrodes #3 and #4 and between #4 and #1, with voltage recorded across electrodes #1 and #2 or #2 and #3. Based on the spatial maps shown in Fig. 2, these current–voltage

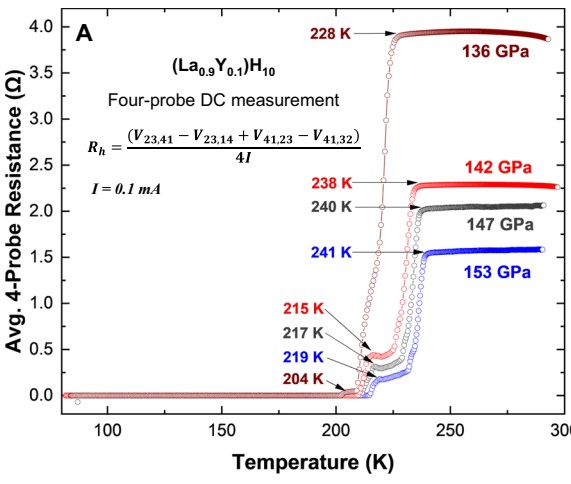

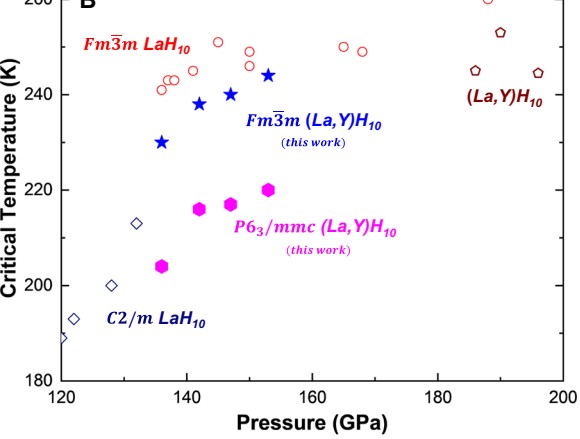

**Fig. 3 | Superconducting behavior of (La,Y)H₁₀ during decompression and comparison with literature. A** Temperature-dependent four-probe DC resistance measurements of (La,Y)H₁₀ collected at multiple pressures during decompression from 153 to 136 GPa, using an excitation current of 0.1 mA. The curves represent the average four-probe resistance, calculated as shown in the inset to eliminate thermoelectric offsets. Distinct drops in resistance indicate superconducting transitions that persist across the entire pressure range. **B** Critical temperature ($T_c$) as a function of pressure for LaH₁₀ and (La,Y)H₁₀ hydrides. Open symbols represent $T_c$ values of LaH₁₀ and (La,Y)H₁₀ reported in the literature[9–12,22]. Solid symbols indicate $T_c$ values of (La,Y)H₁₀ obtained in this work.

pathways intersected domains where the $Fm\bar{3}m$ phase was concentrated, particularly near electrode #2. This spatial correlation supports the assignment of the higher-temperature superconducting transition to the $Fm\bar{3}m$ clathrate phase.

In contrast, other partial resistance configurations, such as $R_{12,34}$, $R_{12,43}$, $R_{23,41}$, and $R_{23,14}$ (Fig. 4A, C), exhibited broader, two-step transitions, with onsets near 241 K and 218 K, respectively. These configurations passed through regions where both $Fm\bar{3}m$ and $P6_3/mmc$ phases were present, with a higher fraction of the hexagonal phase observed between electrodes #1 and #4. The lower-temperature transition was thus attributed to the $P6_3/mmc$ clathrate phase. This interpretation is consistent with previously reported superconducting transition temperatures in undoped and Y-substituted LaH₁₀ systems (Fig. 3B)[9,10,22]. Notably, both transitions occurred at lower onset temperatures than in pure LaH₁₀, where $T_c$ typically exceeds 250 K under similar pressures[9–11]. The observed $T_c$ suppression provides complementary evidence of successful Y incorporation and its influence on the electronic structure, particularly through added intermediate-frequency phonon modes[15,22]. Furthermore, the direct correlation between partial resistances and spatial phase distribution underscores the utility of structural mapping for interpreting superconducting transport behavior in mixed-phase systems.

To evaluate the pressure dependence of superconductivity in the coexisting clathrate phases of (La₀.₉Y₀.₁)H₁₀, resistance measurements were performed during decompression from 153 to 136 GPa (Fig. 3A). Across this pressure range, the resistance–temperature profiles consistently displayed two distinct superconducting transitions, indicative of phase coexistence. Additional support for superconductivity comes from current–voltage (I–V) curve measurements at 146 and 136 GPa (Fig. S11), which exhibit nonlinear behavior below ~230 K at 146 GPa. Notably, the overall resistance behavior and partial resistance traces remained similar down to 142 GPa (Fig. S12). Consistent with measurements at higher pressure, all transition temperatures at each pressure during decompression occurred within experimental uncertainty across the eight configurations, confirming that the observed features are intrinsic to the sample. At 136 GPa, a marked departure from the previous $R_h$–T profiles were observed (Fig. 3). The two-step superconducting features became less distinct, and the transition width narrowed ($\Delta T$ ~20 K), particularly in partial resistances such as $R_{12,34}$, $R_{12,43}$, $R_{23,41}$, and $R_{23,14}$ (Fig. S13). These current paths intersect regions where the spatial phase map (Fig. 2) indicated reduced $Fm\bar{3}m$

intensity near Pt lead #4, consistent with diminished cubic phase contributions at lower pressures.

Meanwhile, traces such as $R_{34,12}$, $R_{34,21}$, $R_{41,23}$, and $R_{41,32}$ still exhibited sharp resistance drops, with more pronounced secondary features. These traces traverse regions around Pt leads #2 and #3, where the $Fm\bar{3}m$ phase remained spatially concentrated even after decompression. The persistence of sharp transitions in these configurations suggests that residual cubic domains retain superconductivity near 228 K, albeit with reduced volume fraction. Structural data at 136 GPa revealed minor distortions in Bragg peak positions of the $Fm\bar{3}m$ phase (Fig. S8), as previously discussed in the structural characterization section. These distortions likely reflect lattice relaxation and non-uniform pressure gradients, and correlate with the observed suppression of the higher-$T_c$ onset from 238 K at 142 GPa to 228 K at 136 GPa, a more rapid decline than typically reported in binary LaH₁₀ systems[9,10,12]. While pure LaH₁₀ transitions to lower-symmetry $C2/m$ or $R\bar{3}m$ phases near this pressure, no such transformations were evident here[6,12], suggesting that the observed $T_c$ suppression arises from phase dilution, lattice strain, and microstructural inhomogeneity, mechanisms known to influence superconductivity in clathrate hydrides[9,10,12,38].

## Discussion

In this study, we demonstrated that partial Y substitution in LaH₁₀ enables the synthesis and stabilization of coexisting cubic $Fm\bar{3}m$ and hexagonal $P6_3/mmc$ clathrate phases in (La₀.₉Y₀.₁)H₁₀ across a wide pressure range, extending down to 136 GPa, well below the structural stability limit of pure LaH₁₀. Using synchrotron-based SXDM and XDI, we directly mapped μm-scale domain heterogeneity and visualized the spatial distribution of clathrate phases across the sample. By integrating these structural maps with multi-channel four-probe resistance measurements, we identified two distinct superconducting transitions: one near 244 K associated with $Fm\bar{3}m$-rich domains, and another near 220 K linked to $P6_3/mmc$-dominated regions. This spatially resolved correlation highlights how microscopic phase separation governs the macroscopic transport response in ternary superhydrides.

Our findings show that Y incorporation extends the structural stability of LaH₁₀-type clathrates without inducing secondary phases or substantially degrading the superconducting critical temperature. Furthermore, the combination of high-resolution structural imaging and spatially sensitive transport measurements provides a robust

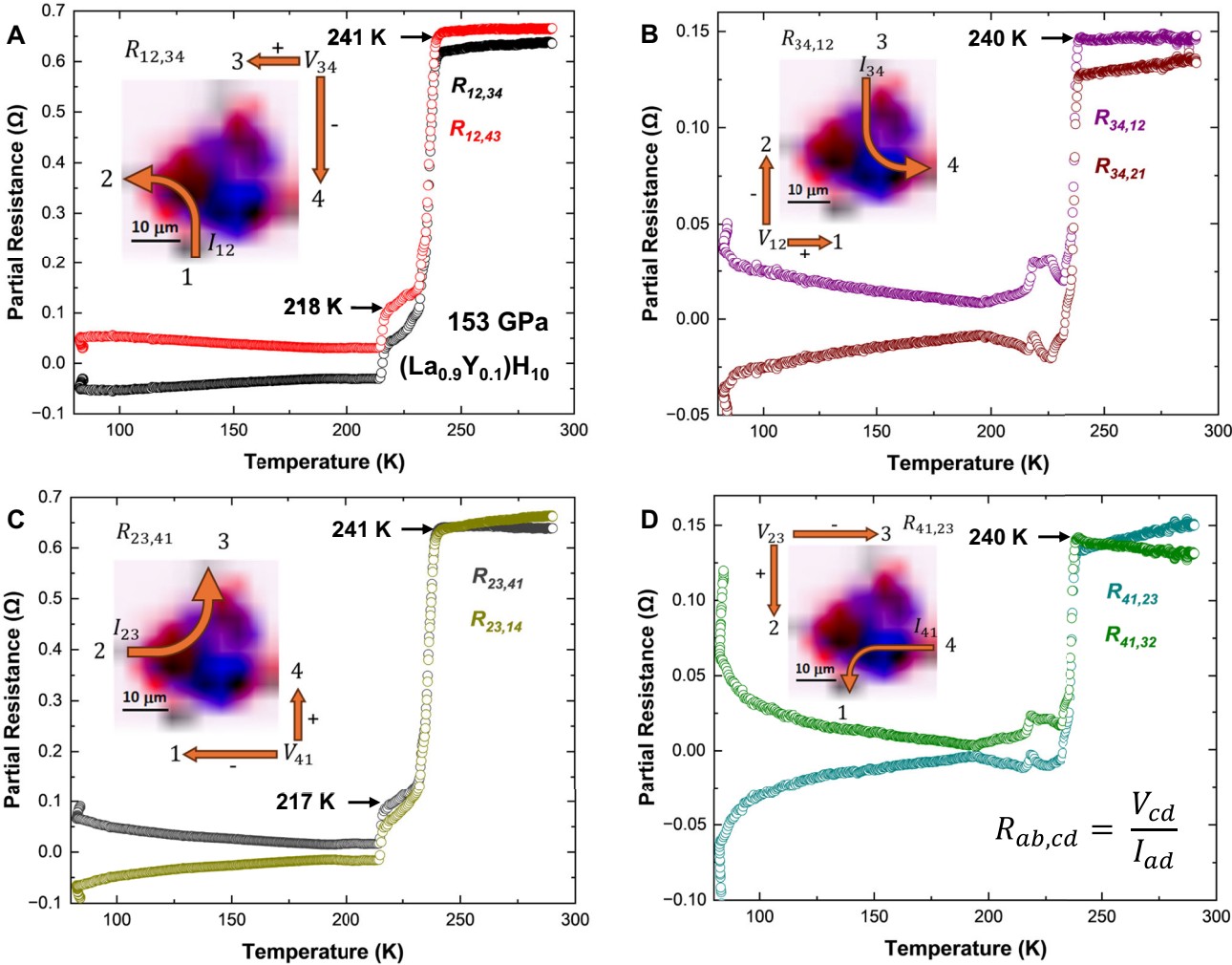

**Fig. 4 | Spatial correlation between superconducting transitions and structural domains in (La$_{0.9}$Y$_{0.1}$)H$_{10}$ at 153 GPa. A–D** Temperature-dependent four-probe partial resistance traces ($R_{ab,cd}$) measured using eight current–voltage configurations are grouped into four panels corresponding to distinct current paths: **A** $R_{12,34}$ and $R_{12,43}$; **B** $R_{34,12}$ and $R_{43,21}$; **C** $R_{23,41}$ and $R_{23,14}$; **D** $R_{41,23}$ and $R_{41,32}$. Each plot includes a schematic of the corresponding current path overlaid on the composite XDI map, highlighting the spatial distribution of cubic $Fm\bar{3}m$ (red) and hexagonal $P6_3/mmc$ (blue) phases. Electrode positions and current directions are annotated. Configurations that differ only by voltage polarity (e.g., $R_{34,12}$ vs. $R_{34,21}$) are spatially equivalent; for clarity, only one representative XDI map is shown for each pair.

Partial resistance traces corresponding to current paths intersecting FCC-enriched domains (e.g., $R_{34,12}$, $R_{41,23}$) exhibit sharp superconducting transitions near 240 K with narrow widths ($\Delta T < 10\,\text{K}$). In contrast, broader or multi-step transitions are observed in configurations that sample mixed-phase or HCP-dominated regions (e.g., $R_{12,34}$, $R_{23,14}$), with onsets near 241 and 218 K. We note that due to single permutations inability to account for voltage drops due to the Seebeck effect, partial resistances exhibit negative values below $T_c$[49]. Averaging reverse-biased polarities corrects for this instrumental artifact, causing an apparent drop to zero resistance. These observations establish a direct spatial correlation between local structural heterogeneity and superconducting behavior.

framework for probing electronic heterogeneity in high-pressure hydrides. More broadly, this work establishes a methodology for probing phase-separated superconductors with µm-scale resolution under extreme conditions. By bridging structural imaging and transport diagnostics, this approach opens new pathways for understanding emergent electronic phenomena in chemically substituted hydrides and informs design strategies aimed at optimizing superconductivity in multicomponent clathrate systems. An exciting next step would also be combining these techniques with new Meissner-effect imaging methods[39]. The successful stabilization and correlation of high-$T_c$ superconductivity in such complex materials mark a step toward the realization of practical hydrogen-based superconductors.

## Methods
### Sample preparation and high-pressure synthesis
A La–Y alloy with a nominal composition of La$_{0.9}$Y$_{0.1}$ was obtained from Ames Laboratory, prepared by arc melting high-purity La (93.36%) and

Y (6.64%) metals under an argon (Ar) atmosphere[40]. Scanning electron microscopy–Energy-dispersive X-ray spectroscopy (SEM–EDS) analysis at 5 µm resolution confirmed a uniform single-phase alloy with a measured composition of 92.33% La and 7.67% Y (Table S1, Fig. S14). Ambient-pressure XRD likewise showed no evidence of phase segregation prior to loading (Fig. S15), confirming that the alloy was a homogeneous precursor. DACs equipped with double-beveled diamonds (-65 µm culet) were used for high-pressure synthesis and subsequent characterization. DAC #1 was configured for transport measurements: the Re gasket was electrically insulated from the sample by prefilling the chamber with a compressed cubic boron nitride (cBN) and epoxy mixture, followed by laser drilling to form the sample chamber. Thin platinum foils (-2 µm thick) were manually positioned on the diamond culet to form electrical contacts in a VDP geometry for four-probe DC resistance measurements[30], with secondary connections made externally using fine copper wires and silver epoxy. DAC #2 employed a Re gasket for synthesis experiments.

Optical images of the sample chambers for both DACs are shown in Fig. S2, and a schematic of the DAC assembly is provided in Fig. S3.

Sample loading was performed in an argon-filled glovebox to avoid air exposure. A piece of the La−Y alloy was placed in direct contact with ammonia borane, which acted as both the hydrogen source and the pressure-transmitting medium[9,41]. After sealing, the DACs were gradually compressed to the target synthesis pressures in multiple steps, with XRD measurements collected at intermediate pressures. Pressure was determined from the Raman shift of the diamond edge[42,43], representative spectra at different pressures are shown in Fig. S16. In situ laser heating was performed at 158 GPa on DAC #1 at beamline 16-ID-B of the APS-U; this sample was subsequently used for both XDI and transport measurements (Fig. 1). A second synthesis was carried out at 172 GPa using DAC #2 at beamline 13-ID-D of the APS, with results shown in Fig. S1. Laser heating in both cases was performed at the interface between the metal and ammonia borane using a modulated Yb fiber laser, delivering ~300 ms focused pulses, reaching estimated temperatures of 1200−1800 K[44,45]. Each synthesis involved multiple laser heating cycles, which ensured complete hydrogen uptake and phase formation.

### Synchrotron X-ray diffraction and imaging

Synchrotron XRD measurements were performed at beamline 16-ID-B of the APS-U (DAC #1, $\lambda = 0.4859$ Å at 158 GPa on the LH table and $\lambda = 0.4246$ Å at 136 GPa on the GP table) and at beamline 13-ID-D of the APS (DAC #2, $\lambda = 0.3344$ Å) at Argonne National Laboratory. Diffraction patterns were collected using PILATUS 2D area detectors and integrated into one-dimensional intensity profiles using DIOPTAS[46]. Structural parameters were extracted via Le Bail refinements using Jana2006[47,48]. This approach was necessary due to spotty diffraction and overlapping contributions from multiple coexisting phases, some with partially known or unknown structural models.

At 13-ID-D, the pre-upgrade X-ray beam was focused to ~3.8 × 2.7 μm². At 16-ID-B, the beam profile improved significantly with the APS-U upgrade: the beam size was reduced from ~6.3 × 3.1 μm² (pre-upgrade) to ~1.4 × 1.2 μm² (post-upgrade), with enhanced brilliance and minimal tails. This improvement enabled high-resolution spatial mapping with minimal signal overlapping between adjacent positions.

To visualize spatial phase distributions and structural inhomogeneity, SXDM was performed at 16-ID-B by raster scanning the laser-heated region over a two-dimensional grid. Each scan consisted of an 11 × 11 array of diffraction patterns acquired with step sizes of 2–5 μm. The collected datasets were processed using the X-ray Diffraction Imaging (XDI) software[26]. For each scanned point, XDI converts the 2D diffraction pattern into a radial intensity profile and integrates the signal within user-defined 2θ regions of interest corresponding to specific Bragg reflections. The resulting integrated intensities were compiled into two-dimensional maps, where each pixel represents a scanned position. The color intensity in these maps reflects the relative strength of the selected reflection, providing a direct visualization of local domain structure, spatial phase distribution, and structural heterogeneity[26,36].

### Electrical transport measurements

After synthesis and XRD analysis, DAC #1 was transferred to a continuous-flow cryostat for temperature-dependent resistance measurements down to 80 K. Electrical resistance measurements were carried out at 153, 147, 142, and 136 GPa. Data was collected during both cooling and warming cycles; however, only the warming cycle (at ~1 K/min) was analyzed due to reduced thermal gradients upon the slow warmup reducing temperature uncertainty. Resistance measurements were performed using a Keithley 6220 current source (100 μA excitation), a 2182A nanovoltmeter, and a Keithley 7001 switching matrix to acquire voltages across multiple electrode configurations[49]. Current and voltage leads were permuted in the standard VDP configuration[30]. Measured partial resistances are then

defined as $R_{ab,cd} = \frac{V_{cd}}{I_{ab}}$, where $I_{ab}$ is the applied current between contacts $a$ and $b$, and $V_{cd}$ being the voltage measured across contacts $c$ and $d$. Because the sample did not satisfy the assumptions required for extracting sheet resistivity ($R_s$) using the standard VDP equation,

$$e^{\frac{-\pi\left(R_{12,34}+R_{34,12}\right)}{2R_s}} + e^{\frac{-\pi\left(R_{23,41}+R_{41,23}\right)}{2R_s}} = 1,$$

Primarily due to sample inhomogeneity, the resultant $R_s$ values are not expected to represent the bulk electronic properties of a single phase. Nonetheless, VDP-averaged resistances are reported, as the superconducting transition is clearly observed (Fig. 3A). Correlations between individual partial resistances and the spatial phase distribution are explored in this work.

## Data availability
All relevant data are available from the corresponding authors upon request.

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

## Acknowledgements

We thank J. S. Schilling, J. H. Eggert, and A. F. Abdulla for helpful comments and discussions. This work was supported by the U.S. Department of Energy (DOE)-National Nuclear Security Administration (NNSA; DE-NA0004153, Chicago/DOE Alliance Center), National Science Foundation (NSF, DMR-2104881), and DOE-Office of Science (DE-SC0020340), to A.H.M.M., K.W., N.S., M.A., A.C.M., and R.J.H. Operations of HPCAT (Sector 16, APS, ANL) are supported by DOE-NNSA Office of Experimental Sciences. Operations of GeoSoilEnviroCARS (University of Chicago, Sector 13, APS, ANL) are supported by NSF Earth Sciences program via SEES: Synchrotron Earth and Environmental Science (EAR—2223273). Work at the Advanced Photon Source was supported by the U.S. DOE Office of Science, Office of Basic Energy Sciences, under Award No. DE-AC02-06CH11357.

## Author contributions

A.H.M.M. and R.J.H. designed the research. A.H.M.M., K.W., N.P.S., and M.A. prepared the experiments. A.H.M.M., K.W., N.P.S., M.A., R.H., S.C., D.S., V.B.P., M.S., and N.V. performed synchrotron-based experiments. A.H.M.M., A.C.M., K.W., N.P.S., M.A., and R.J.H. conducted the four-probe electrical transport measurements. A.H.M.M., K.W., N.P.S, A.C.M., M.A., and R.J.H. analyzed data. All authors participated in discussing the results and writing the paper.

## Competing interests

The authors declare no competing interests.
