## [Transparent Peer Review file · Nature Communications]

X-ray Diffraction and Electrical Transport Imaging of Superconducting Superhydride (La,Y)H₁₀

Corresponding Author: Mr Abdul Haseeb Manayil Marathamkottil

Version 0:

Reviewer comments:

Reviewer #1

(Remarks to the Author)

Please see the attached pdf version.

Reviewer #2

(Remarks to the Author)

The manuscript reports (La_{0.9}Y_{0.1})H₁₀ synthesized ≥ 150 GPa with coexisting cubic and hexagonal clathrate phases from 172 \rightarrow 136 GPa, mapped by SXDM/XDI at APS-U with ~ 1 μ m beam, and correlated to eight van der Pauw permutations showing two superconducting onsets (~ 244 K and ~ 220 K) that spatially associate with FCC-rich vs HCP-rich domains. The core contribution is the spatial co-analysis linking local structure to transport signatures in a mixed-phase superhydride—compelling and timely.

A few minor comments need to be addressed before final acceptance.

1. To evaluate the homogeneity of the as-cast La-Y ingot, specify the grain size of the SEM-EDS in Table S1. How can you confirm the uniformity of the alloy, considering your sample size, ~ 30 μ m? Put a larger font of the scale bar in Fig. S2.
 2. Evaluate the actual stoichiometry of FCC and hcp phases after laser heating.
 3. Provide the Raman data demonstrating surplus hydrogen remained after LH.
 4. Clarify pressure gradients across the chamber (e.g., diamond edge Raman at multiple azimuths) and map how they correlate with domain distribution.
- end.

Version 1:

Reviewer comments:

Reviewer #1

(Remarks to the Author)

I can accept the publication of the article and have no further suggestions

Reviewer #2

(Remarks to the Author)

The authors have sufficiently addressed my concerns. I have no additional comments and recommend this manuscript for publication in Nature Communications.

X-ray Diffraction and Electrical Transport Imaging of Superconducting Superhydride (La,Y)H₁₀

A. H. Manayil Marathamkottil^{1*}, Kui Wang², Nilesh P. Salke², Muhtar Ahart², Alexander C. Mark², Ross Hrubciak³, Stella Chariton⁴, Dean Smith³, Vitali B. Prakapenka⁴, Maddury Somayazulu³, Nenad Velisavljevic^{3,5}, Russell J. Hemley^{1,2,6*}

Authors' Response to Reviewer Comments

Reviewer #1:

This paper reports the synthesis and characterization of (LaY)H₁₀ superhydrides. Two coexisting cubic *Fm-3m* and hexagonal *P63/mmc* phases were observed over the studied pressure range. The μm -scale structural inhomogeneity was revealed and visualized via SXDM and XDI. Four-probe DC resistance measurements confirmed superconductivity with two distinct transitions. The first T_c is 244 K associated with the cubic phase and the second T_c near 220 K linked to the hexagonal phase. However, a few points need clarification before it can be recommended for acceptance.

Response:

We sincerely thank the reviewer for their thoughtful and encouraging summary of our work. We are especially grateful that they highlighted the synthesis, phase coexistence, and superconducting transitions in (La,Y)H₁₀, as well as the observation of μm -scale structural inhomogeneity via SXDM/XDI and the identification of two distinct superconducting transitions from four-probe resistance measurements. We truly appreciate the constructive suggestions and have carefully addressed each point in detail below.

Comment #1:

Why the number and shape of the XRD diffraction peaks are different at 158 GPa and 172 GPa for $\text{La}_{0.9}\text{Y}_{0.1}$? such as the peaks at 7, 10, and 13.5 degrees at 158 GPa.

Response:

We carefully compared the $\text{La}_{0.9}\text{Y}_{0.1}$ diffraction patterns at 158 and 172 GPa and revised the manuscript to clarify differences between the number and shape of the diffraction peaks. Three main factors contribute: (i) measurement geometry, since the 172 GPa data were collected at $\lambda = 0.3344 \text{ \AA}$ at Sector 13-IDD of GSECARs and the 158 GPa data at $\lambda = 0.4859 \text{ \AA}$ at Sector 16-IDB of HPCAT, resulting in different Q-ranges; (ii) microstructure, with the 172 GPa sample producing uniform powder rings, while the 158 GPa sample showed textured arcs that redistributed intensity; and (iii) sample environment, as the 158 GPa transport cell contained Pt electrodes, NH_3BH_3 and a c-BN+epoxy gasket, whose reflections overlapped with weak La peaks. These effects explain why fewer La reflections are visible at 158 GPa.

We also clarified that peak-shape differences, such as broadening of the (002) reflection, are due to pressure-induced distortions and texture, which are well documented in high-pressure La

studies^{1,2}. Finally, we explicitly labeled Pt peaks and marked the weak features near 7° and 10° as unknown impurity reflections to avoid confusion.

Changes in the manuscript:

- **From Results – Structural Characterization of (La,Y)H₁₀**

“At 158 GPa (DAC #1, Fig. 1), the diffraction pattern showed spotty diffraction rings, a narrower accessible 2θ range at the longer wavelength, and additional peaks from Pt electrodes and the sample-environment, which obscured weaker La-Y reflections. By contrast, at 172 GPa (DAC #2, Fig. S1), the shorter wavelength, absence of electrode and gasket contributions, and improved powder averaging yielded well-resolved La-Y reflections. A direct comparison of La_{0.9}Y_{0.1} at 158 and 172 GPa is provided in Fig. S2, showing that the observed differences in the number and shape of peaks arise from experimental factors rather than intrinsic structural changes. In addition, the broader and asymmetric (002) profile at 172 GPa reflects pressure-induced lattice distortions and deviatoric stress, consistent with prior observations in elemental La under compression.”

- **From Methods – Sample Preparation and High-Pressure Synthesis**

“DAC #1 was configured for transport measurements: the Re gasket was electrically insulated from the sample by prefilling the chamber with a compressed cubic boron nitride (cBN) and epoxy mixture, followed by laser drilling to form the sample chamber. Thin platinum foils (~2 μm thick) were manually positioned on the diamond culet to form electrical contacts in a VDP geometry for four-probe DC resistance measurements³, with secondary connections made externally using fine copper wires and silver epoxy. DAC #2 employed a Re gasket for synthesis experiments. Optical images of the sample chambers for both DACs are shown in Fig. S2, and a schematic of the DAC assembly is provided in Fig. S3.”

- **From Methods – Synchrotron X-ray Diffraction and Imaging**

“Synchrotron XRD measurements were performed at beamline 16-ID-B of the APS-U (DAC #1, λ = 0.4859 Å at 158 GPa on the LH table and λ = 0.4246 Å at 136 GPa on the GP table) and at beamline 13-ID-D of the APS (DAC #2, λ = 0.3344 Å) at Argonne National Laboratory.”

- **From Figure 1 - XRD patterns and structural refinement of coexisting (La,Y)H₁₀ phases**

Figure 1: Bottom Panel: Experimental XRD patterns of $\text{La}_{0.9}\text{Y}_{0.1}$ at 158 GPa before and after laser heating. The pre-heating pattern corresponds to the distorted-cubic $Fm\bar{m}m$ phase. After laser heating, the diffraction pattern shows the formation of $(\text{La},\text{Y})\text{H}_{10}$ phases, consistent with the refined structures shown above. Reflections from Pt electrodes are explicitly labeled, while peaks marked with “*” correspond to unidentified or sample-environment contributions. Insets show representative 2D diffraction images for reference.

- From Figure S2 - Comparison of $\text{La}_{0.9}\text{Y}_{0.1}$ samples at 158 and 172 GPa

Figure S2: Comparison of $\text{La}_{0.9}\text{Y}_{0.1}$ samples at 158 and 172 GPa. Left: Optical images of the loaded chambers in DAC #1 and DAC #2, illustrating differences in sample environment. Center: 2D diffraction images, showing spotty rings at 158 GPa and uniform powder rings at 172 GPa. Right: Integrated XRD patterns. At 158 GPa (DAC #1), additional reflections from the cell assembly are present: peaks at $d \approx 1.79$ and 2.2 \AA correspond to fcc-Pt, while broad features at $d \approx 2.80$ and 4.0 \AA are marked as impurity and likely arise from amorphous gasket contributions. At 172 GPa (DAC #2), the pattern is cleaner and dominated by La–Y reflections.

Comment #2:

Although the author used the Le Bail method for fitting, several peaks remain unknown, as shown in the figure below. Moreover, the fitting deviation for the 102 peak is excessively large. Could these peaks originate from a new structural phase? Additionally, the authors should provide the XRD diffraction pattern alongside the integrated intensity plot.

Response:

We carefully re-analyzed the diffraction data to address these points. The revised Le Bail refinements now explicitly account for the reflections near 13.5° and 15.5° , which originate from fcc-Pt electrodes. Other very weak reflections remain unidentified and are labeled as impurity peaks from sample-environment in the updated figures. Their intensities are too low and isolated to support assignment to a new structural phase. Similar minor impurity features have been reported in previous studies of La-H and La-Y-H superhydrides^{4,5}.

The apparent deviation in the 102 reflection is now minimized after re-refinement. Residual mismatches at this pressure (153 GPa) are consistent with nonhydrostatic stress and lattice distortions, which are known to broaden or slightly shift reflections in lanthanum under megabar conditions¹. This does not indicate the presence of a new phase.

To increase transparency, we updated the manuscript by (i) labeling Pt and impurity peaks in the Le Bail fits, and (ii) providing corresponding 2D diffraction images alongside the integrated intensity plots so that azimuthal distributions can be compared directly.

Changes in the manuscript:

- From Figure 1 - XRD patterns and structural refinement of coexisting (La,Y)H₁₀ phases

Figure 1: Top: XRD patterns and structural refinement of coexisting (La,Y)H₁₀ phases. Top: Experimental synchrotron XRD patterns and Le Bail refinements of the *Fm*-3*m* and *P6*₃/*mmc* (La,Y)H₁₀ phases at 153 GPa. The experimental data, fit, and residuals are shown in red, black, and gray, respectively. Refined lattice parameters for both phases are indicated. The peaks marked with "*" correspond to unidentified or sample-environment contributions. Insets show representative 2D diffraction images for reference.

Comment #3:

I suggest changing "DAC#2" to "DAC#1" in the main text to enhance clarity for readers.

Response:

We revised the Methods section to clarify the DAC numbering and improve readability. DAC #1 is now described as the transport cell, and DAC #2 as the synthesis-focused cell, with the corresponding beamline and wavelength details specified.

Changes in the manuscript include:

- **From Methods – Sample Preparation and High-Pressure Synthesis**

“DAC #1 was configured for transport measurements: the Re gasket was electrically insulated from the sample by prefilling the chamber with a compressed cubic boron nitride (cBN) and epoxy mixture, followed by laser drilling to form the sample chamber. Thin platinum foils (~2 μm thick) were manually positioned on the diamond culet to form electrical contacts in a VDP geometry for four-probe DC resistance measurements, with secondary connections made externally using fine copper wires and silver epoxy. DAC #2 employed a Re gasket for synthesis experiments.”

- **From Methods – Synchrotron X-ray Diffraction and Imaging**

“Synchrotron XRD measurements were performed at beamline 16-ID-B of the APS-U (DAC #1, $\lambda = 0.4859 \text{ \AA}$ at 158 GPa on the LH table and $\lambda = 0.4246 \text{ \AA}$ at 136 GPa on the GP table) and at beamline 13-ID-D of the APS (DAC #2, $\lambda = 0.3344 \text{ \AA}$) at Argonne National Laboratory.”

Comment #4:

Why do the positions of the unknown peaks differ between 153 GPa and 136 GPa in DAC#2? What is the origin of these unknown peaks? Were the XRD patterns collected from the same sample location for these two measurements? The XRD diffraction pattern and micrograph of the collection positions should be provided.

Response:

We revised the manuscript to clarify the origin of these peaks and added new supplementary figures to show the measurement locations. Figure S5 now presents the XRD collection positions at 153 GPa and 136 GPa in DAC #1, corresponding to the patterns in Fig. 1 and Fig. S6. Figure S7 shows paired XRD patterns at 156 GPa and 136 GPa collected from equivalent positions, confirming the coexistence of cubic and hexagonal phases at both pressures. Figure S8 provides diffraction images from different regions at 136 GPa, highlighting distortions in the fcc lattice upon decompression.

The apparent differences between the 153 GPa and 136 GPa patterns are explained by:

- Sampling location: the 153 GPa dataset was collected near the electrode region, where Pt reflections appear, whereas the 136 GPa dataset was collected at the sample center, with negligible Pt contribution.
- Stress-related distortions: decompression leads to distortions in the fcc (La,Y)H₁₀ phase, producing weak features as seen in Fig. S8.

- Instrument configuration: the 153 GPa pattern was collected on the LH table with $\lambda = 0.4859$ Å, while the 136 GPa pattern was measured in a separate run on the GP table with $\lambda = 0.4246$ Å. This difference naturally shifts 2θ values even when d-spacings remain unchanged.

To remove ambiguity, we also included a comparison in d-spacing (Fig. S7 Right), which shows that all observed shifts are consistent with decompression and do not indicate a new phase.

Changes in the manuscript:

- From Results – Structural Characterization of (La,Y)H₁₀**

“To assess the stability of the observed phases during decompression, additional XRD measurements were performed on both samples. In DAC #1, the sample was gradually decompressed from 153 GPa, and an XRD was collected at 136 GPa from the sample center, with Fig. S5 illustrating the collection positions corresponding to the patterns in Fig. 1 and Fig. S6. Electrical resistance measurements were carried out in the intermediate pressure range. At 136 GPa, both the cubic and hexagonal phases were still observed (Fig. S6 and S7). Figure S7 presents XRD patterns of (La,Y)H₁₀ at 153 GPa and 136 GPa in DAC #1, collected from comparable positions in the sample chamber to confirm the persistence of both phases across this pressure range. At 136 GPa, minor distortions in the $Fm\bar{3}m$ phase were also evident across different regions of the sample (Fig. S8), likely reflecting local pressure gradients and lattice relaxation.”

- From Figure S5 - XRD collection positions for (La,Y)H₁₀ at 153 GPa and 136 GPa**

Figure S5: XRD collection positions for (La,Y)H₁₀ at 153 GPa and 136 GPa in DAC #1, corresponding to the patterns shown in Fig. 1 and Fig. S5. **Left:** XDI map of the sample chamber

at 153 GPa and 2D scan of the chamber at 136 GPa, with electrodes and sampling locations marked for spatial reference. **Right:** representative 2D diffraction images collected at the indicated locations. The 153 GPa pattern was collected near the electrode region, where Pt peaks are visible, while the 136 GPa pattern was collected at the sample center, where Pt contributions are within the signal-to-noise ratio.

- From Figure S7 - XRD patterns of $(\text{La},\text{Y})\text{H}_{10}$ at 153 GPa and 136 GPa in DAC #1

Figure S7: XRD patterns of $(\text{La},\text{Y})\text{H}_{10}$ at 153 GPa and 136 GPa in DAC #1, collected from comparable positions in the sample chamber to confirm the presence of cubic and hexagonal phases at both pressures. The data were taken from the central square (55th and 56th positions in a 110-square grid) to ensure consistent sampling. **Left:** XDI maps of the sample chamber. **Middle:** representative 2D diffraction images from the selected positions. **Right:** integrated diffraction patterns in d-spacing, demonstrating that the two datasets are directly comparable.

- From Figure S8 - Spatially resolved XRD analysis of the cubic phase at 136 GPa

Figure 2: Spatially resolved XRD analysis of the cubic phase at 136 GPa. XDI map of the $\text{Fm}\bar{3}\text{m}$ (cubic) phase of $(\text{La},\text{Y})\text{H}_{10}$ at 136 GPa, showing its spatial distribution across the scanned region.

Diffraction patterns extracted from two high-intensity regions of the grid (right) confirm the presence of the cubic phase with slight peak shifts between locations, likely arising from local pressure gradients and lattice relaxation effects.

Comment #5:

In the main text, the author named the sample as (La_{0.9}Y_{0.1})H₁₀, How did the author arrive at this ratio? And What evidence supports the successful incorporation of Y for the reaction product?

Response:

We clarified in the manuscript how the composition was determined and provided multiple lines of evidence for yttrium incorporation. The La–Y alloy precursor was prepared by arc-melting 90% La and 10% Y at Ames Laboratory. SEM–EDS confirmed a uniform single-phase alloy with 92.33% La and 7.67% Y (Fig. S14 & S15), and ambient XRD showed no segregation.

Following hydrogenation, we performed a volume-based stoichiometry analysis for both phases. For the cubic Fm-3m phase at 153 GPa, the refined unit-cell volume corresponds to ~9.96 H per metal atom. For the hexagonal P6₃/mmc phase, the expansion corresponds to ~9.34 H per metal atom. Both are consistent with decahydride stoichiometry.

Evidence of Y incorporation includes:

- Slightly contracted unit-cell volumes relative to LaH₁₀ at the same pressure (Fig. S4), consistent with substitution by the smaller Y atom.
- Persistence of both cubic and hexagonal clathrate phases down to 136 GPa, whereas undoped LaH₁₀ typically transforms to lower-symmetry phases.
- A measured reduction in T_c compared to LaH₁₀, consistent with prior reports that Y lowers T_c while stabilizing the clathrate lattice⁵.

We also added a new *Supplementary Note 1* detailing the stoichiometry analysis.

Changes in the manuscript:

- **From Results – Structural Characterization of (La,Y)H₁₀**

“As shown in Fig. S4, this systematic reduction provides direct structural evidence that ~10% Y is incorporated into the clathrate framework. Notably, no diffraction peaks from secondary LaH_n or YH_n phases were observed, indicating that Y substitution remains within the solubility limit for

forming a single-phase clathrate or a mixed-phase clathrate solid solution. Volume-based stoichiometry analysis yielded ~ 10 H per metal atom for both the cubic and hexagonal phases, consistent with the nominal $(\text{La}_{0.9}\text{Y}_{0.1})\text{H}_{10}$ composition (Supplementary Note 1).”

- **From Results – Structural Characterization of $(\text{La},\text{Y})\text{H}_{10}$**

“In DAC #2, after initial synthesis at 172 GPa and characterization at 168 GPa, the sample was decompressed to 161 GPa. At this pressure, both the $Fm\bar{3}m$ and $P6_3/mmc$ phases clearly present (Fig. S4), in contrast to binary La hydrides where the cubic phase typically dominates. The coexistence of both clathrate phases at this stage of decompression underscores the role of Y substitution in stabilizing structural polymorphism beyond that observed in undoped LaH_{10} . The consistent observation of both phases across the entire pressure range studied (172–136 GPa) confirms that they coexist rather than represent distinct, pressure-stabilized states. Their simultaneous presence indicates that Y substitution does not strongly favor one structure over the other but instead promotes a mixed-phase state. This contrasts with LaH_{10} , which transforms to lower-symmetry structures upon decompression.”

- **From Results – Structural Characterization of $(\text{La},\text{Y})\text{H}_{10}$**

“The pressure–volume (P – V) behavior of $(\text{La}_{0.9}\text{Y}_{0.1})\text{H}_{10}$, shown in Fig. S4, generally follows the trend reported for undoped LaH_{10} , with modest phase-specific deviations that point to non-uniform compressibility and local strain effects. Similar trends were reported for $(\text{La}_{0.8}\text{Y}_{0.2})\text{H}_{10}$ synthesized at higher pressures⁵, where clathrate structures remained stable without decomposition. These results demonstrate that partial Y substitution preserves the hydrogen cage framework of LaH_{10} while extending phase coexistence across a broader pressure range.”

- **From Methods – Sample Preparation and High-Pressure Synthesis**

“A La–Y alloy with a nominal composition of $\text{La}_{0.9}\text{Y}_{0.1}$ was obtained from Ames Laboratory, prepared by arc melting high-purity La (93.36%) and Y (6.64%) metals under an argon (Ar) atmosphere⁴⁰. Scanning Electron Microscopy - Energy-Dispersive X-ray Spectroscopy (SEM–EDS) analysis at 5 μm resolution confirmed a uniform single-phase alloy with a measured composition of 92.33% La and 7.67% Y (Table S1, Fig. S14). Ambient-pressure XRD likewise showed no evidence of phase segregation prior to loading (Fig. S15), confirming that the alloy was a homogeneous precursor.”

- **From Supplementary Information Note 1 – Volume-Based Stoichiometry Analysis of $(\text{La},\text{Y})\text{H}_{10}$**

“To estimate the hydrogen content of the synthesized clathrate phases, we compared the atomic volumes of the La–Y alloy precursor and the hydrogenated phases, following procedures established for rare-earth hydrides. All values reported here were obtained from DAC #1 (Fig. 1). The alloy measurement was performed at 158 GPa, while the hydride phases were measured at 153 GPa; the ± 5 GPa difference is within the experimental uncertainty of pressure determination and therefore allows direct comparison.

- $\text{La}_{0.9}\text{Y}_{0.1}$ alloy precursor ($Fmmm$, 158 GPa):

Unit cell volume = 58.5 Å³ with 4 metal atoms → atomic volume = 14.63 Å³/atom.

- Cubic $Fm\bar{3}m$ phase (153 GPa):

Unit cell volume = 136.6 Å³ with 4 formula units → atomic volume = 34.15 Å³/metal atom.

Expansion relative to alloy = 19.52 Å³/atom.

- Hexagonal $P6_3/mmc$ phase (153 GPa):

Unit cell volume = 66.1 Å³ with 2 metal atoms → atomic volume = 33.05 Å³/metal atom.

Expansion relative to alloy = 18.42 Å³/atom.

- Hydrogen reference:

The atomic volume of hydrogen at 153 GPa is taken as **1.961 Å³/atom**, derived from the Vinet EOS of H₂, using parameters: B₀=0.162 GPa, B₀'=6.813, V₀=21.116 Å³⁶.

From these values, the estimated hydrogen content per metal atom is:

$$n_H = \frac{\Delta V}{V_H}$$

- Cubic phase: 19.52 / 1.96 ≈ 9.96 H/metal atom
- Hexagonal phase: 18.42 / 1.96 ≈ 9.34 H/metal atom

These results indicate that both phases are consistent with decahydride stoichiometry, within experimental uncertainty, and support the assignment of the composition as (La_{0.9}Y_{0.1})H₁₀. This conclusion is further supported by the pressure–volume trends shown in Figure S4, where the data for both FCC and HCP phases closely follow the reported equation of state of LaH₁₀, consistent with ~10 H per metal atom.”

Comment #7:

The images in Figure 2 at 153 and 136 GPa should be presented in the same orientation to facilitate direct comparison and interpretation.

Response:

We revised Figure 2 so that the 153 GPa and 136 GPa XDI images are displayed in a nearly common reference frame. This improves direct comparison while retaining the full uncropped raster grids to avoid loss of peripheral features. The difference in raster orientation between runs (due to stage angle/zoom) prevents exact pixel alignment, but the updated figure enhances clarity without compromising completeness of the dataset.

Changes in the manuscript:

- From Figure 2 - Spatially resolved XRD imaging of phase domains in $(\text{La}_{0.9}\text{Y}_{0.1}\text{H}_{10})$ at high pressure

Figure 2: Spatially resolved XRD imaging of phase domains in $(\text{La}_{0.9}\text{Y}_{0.1}\text{H}_{10})$ at high pressure. (A) XDI maps of the laser-heated $(\text{La},\text{Y})\text{H}_{10}$ sample at 153 GPa (top) and 136 GPa (middle and bottom), obtained via raster scanning with a $\sim 1 \mu\text{m}$ micro-focused synchrotron beam. The 30 $\mu\text{m} \times 30 \mu\text{m}$ scan at 153 GPa resolves spatial distributions of cubic $\text{Fm}\bar{3}\text{m}$ (red) and hexagonal $\text{P6}_3/\text{mmc}$ (blue) domains, with Pt leads mapped in dark gray or brown. Electrode positions are

annotated directly on the Pt to enable spatial correlation with electrical transport measurements. The rightmost column overlays all phases to show the composite spatial distribution. At 136 GPa, a 50 μm \times 50 μm scan (middle) and a higher-resolution 15 μm \times 15 μm scan (bottom) show continued coexistence of FCC and HCP domains with spatial variation. (B) 2D X-ray scan overview showing the raster grid layout at respective pressures. (C) Optical image of the sample at 153 GPa after laser heating.

Comment #8:

The Raman spectroscopy data used for pressure calibration should be provided in the supplementary materials.

Response:

We added the requested data to the Supplementary Information. Figure S16 now presents the diamond-edge Raman spectra used for pressure calibration. In each case, spectra were collected at the culet center of the sample, and they confirm the pressure values reported in the manuscript.

Changes in the manuscript:

- **From Methods – Sample Preparation and High-Pressure Synthesis**

”Pressure was determined from the Raman shift of the diamond edge, representative spectra at different pressures are shown in Fig. S16.”

- **From Figure S16 – Diamond-edge Raman spectra for pressure calibration (DAC #1)**

Figure S16: Diamond-edge Raman spectra for pressure calibration (DAC #1). Representative spectra at 158 GPa (before laser heating), 153 GPa (after laser heating), and 136 GPa (during decompression). The diamond edge was measured at the center of the sample.

Reviewer #2:

The manuscript reports (La_{0.9}Y_{0.1})Hf₁₀ synthesized ≥ 150 GPa with coexisting cubic and hexagonal clathrate phases from 172 \rightarrow 136 GPa, mapped by SXDM/XDI at APS-U with ~ 1 μm beam, and correlated to eight van der Pauw permutations showing two superconducting onsets (~ 244 K and ~ 220 K) that spatially associate with FCC-rich vs HCP-rich domains. The core contribution is the spatial co-analysis linking local structure to transport signatures in a mixed-phase superhydride—compelling and timely. A few minor comments need to be addressed before final acceptance.

Response:

We sincerely thank the reviewer for their thoughtful and encouraging assessment of our work. We are especially grateful that the reviewer recognized the significance of correlating spatially resolved structure with superconducting behavior in superhydrides, which we see as the central contribution of this study. We appreciate the constructive suggestions and have carefully addressed each of them below.

Comment #1:

To evaluate the homogeneity of the as-cast La-Y ingot, specify the grain size of the SEM-EDS in Table S1. How can you confirm the uniformity of the alloy, considering your sample size, ~ 30 μm ? Put a larger font of the scale bar in Fig. S2.

Response:

We revised the Supplementary Information to clarify the SEM–EDS analysis and to document alloy homogeneity across multiple length scales. The caption of Table S1 now explicitly states that the listed values correspond to EDS spectra acquired at 5 μm resolution (Fig. S14). SEM images were acquired over a 100 μm field of view, and additional EDS measurements were collected at 25 μm , 10 μm , and 5 μm resolutions, all of which showed consistent La and Y distributions. These results demonstrate compositional uniformity at scales well below the ~ 30 μm DAC sample size.

This conclusion is corroborated by ambient-pressure synchrotron XRD (Fig. S15), which shows a single-phase dhcp alloy with no evidence of segregation, and by high-pressure diffraction prior to hydrogenation, which reveals only a single distorted-*Fmmm* phase above 150 GPa. Together, these

datasets confirm that the alloy precursor is homogeneous across both microscopic and macroscopic scales.

In addition, Figure S3 has been updated with a larger, clearer scale bar as requested.

Changes in the manuscript:

- **From Methods – Sample Preparation and High-Pressure Synthesis**

“A La–Y alloy with a nominal composition of $\text{La}_{0.9}\text{Y}_{0.1}$ was obtained from Ames Laboratory, prepared by arc melting high-purity La (93.36%) and Y (6.64%) metals under an argon (Ar) atmosphere⁴⁰. Scanning Electron Microscopy - Energy-Dispersive X-ray Spectroscopy (SEM–EDS) analysis at 5 μm resolution confirmed a uniform single-phase alloy with a measured composition of 92.33% La and 7.67% Y (Table S1, Fig. S14). Ambient-pressure XRD likewise showed no evidence of phase segregation prior to loading (Fig. S15), confirming that the alloy was a homogeneous precursor.”

- **From Table S1 – Results of SEM-EDS analysis for $\text{La}_{0.9}\text{Y}_{0.1}$ alloy**

Table S1: Results of SEM–EDS analysis for $\text{La}_{0.9}\text{Y}_{0.1}$ alloy. The listed values correspond to EDS spectra acquired at 5 μm resolution (Fig. S14). SEM imaging was performed over a 100 μm field of view. Additional EDS measurements at 25 μm and 10 μm gave consistent results, supporting alloy homogeneity across multiple length scales.

$\text{La}_{0.9}\text{Y}_{0.1}$						
Element	Line Type	Apparent Concentration	k-Ratio	Wt%	Wt% Sigma	Standard Label
Y	L series	1.08	0.01082	7.67	0.15	Y
La	L series	35.12	0.31512	92.33	0.15	LaB_6
Total:				100.00		

- **From Figure S3 – Schematic illustration of the DAC assembly configured for four-probe electrical transport measurements**

Figure S3: Left: Schematic illustration of the DAC assembly configured for four-probe electrical transport measurements. Right: Optical images of the $(\text{La},\text{Y})\text{H}_{10}$ DAC #1 sample at 158 GPa before laser heating (transmitted and reflected light) and after heating at 153 GPa (transmitted light). A clear volume expansion is observed following synthesis, as evidenced by the transition of the initially transparent ammonia borane region to an opaque state on the right side of the culet.

Comment #2:

Evaluate the actual stoichiometry of FCC and hcp phases after laser heating.

Response:

Thank you for this important point. We performed a volume-based stoichiometry analysis for both the cubic and hexagonal clathrate phases, using the alloy precursor and the refined unit-cell volumes after hydrogenation. Both phases are consistent with ~ 10 H per metal atom, i.e., a nominal decahydride stoichiometry. The slightly lower hydrogen count in the hexagonal phase is attributed to small structural distortions or partial hydrogen occupancy, as expected for this metastable structure.

To provide full transparency, we added a new *Supplementary Note 1* that details the calculations, including reference to the hydrogen equation of state.

Changes in the manuscript:

- **From Results – Structural Characterization of $(\text{La},\text{Y})\text{H}_{10}$**

“As shown in Fig. S4, this systematic reduction provides direct structural evidence that $\sim 10\%$ Y is incorporated into the clathrate framework. Notably, no diffraction peaks from secondary LaH_n or YH_n phases were observed, indicating that Y substitution remains within the solubility limit for forming a single-phase clathrate or a mixed-phase clathrate solid solution. Volume-based

stoichiometry analysis yielded ~ 10 H per metal atom for both the cubic and hexagonal phases, consistent with the nominal $(\text{La}_{0.9}\text{Y}_{0.1})\text{H}_{10}$ composition (Supplementary Note 1).”

- **From SI Note 1 – Volume-Based Stoichiometry Analysis of $(\text{La},\text{Y})\text{H}_{10}$**

“To estimate the hydrogen content of the synthesized clathrate phases, we compared the atomic volumes of the La–Y alloy precursor and the hydrogenated phases, following procedures established for rare-earth hydrides. All values reported here were obtained from DAC #1 (Fig. 1). The alloy measurement was performed at 158 GPa, while the hydride phases were measured at 153 GPa; the ± 5 GPa difference is within the experimental uncertainty of pressure determination and therefore allows direct comparison.

- $\text{La}_{0.9}\text{Y}_{0.1}$ alloy precursor ($Fmmm$, 158 GPa):
Unit cell volume = 58.5 \AA^3 with 4 metal atoms \rightarrow atomic volume = $14.63 \text{ \AA}^3/\text{atom}$.
- Cubic $Fm\bar{3}m$ phase (153 GPa):
Unit cell volume = 136.6 \AA^3 with 4 formula units \rightarrow atomic volume = $34.15 \text{ \AA}^3/\text{metal atom}$.
Expansion relative to alloy = $19.52 \text{ \AA}^3/\text{atom}$.
- Hexagonal $P6_3/mmc$ phase (153 GPa):
Unit cell volume = 66.1 \AA^3 with 2 metal atoms \rightarrow atomic volume = $33.05 \text{ \AA}^3/\text{metal atom}$.
Expansion relative to alloy = $18.42 \text{ \AA}^3/\text{atom}$.
- Hydrogen reference:
The atomic volume of hydrogen at 153 GPa is taken as **$1.961 \text{ \AA}^3/\text{atom}$** , derived from the Vinet EOS of H_2 , using parameters: $B_0=0.162 \text{ GPa}$, $B_0'=6.813$, $V_0=21.116 \text{ \AA}^3$.

From these values, the estimated hydrogen content per metal atom is:

$$n_H = \frac{\Delta V}{V_H}$$

- Cubic phase: $19.52 / 1.96 \approx 9.96 \text{ H/metal atom}$
- Hexagonal phase: $18.42 / 1.96 \approx 9.34 \text{ H/metal atom}$

These results indicate that both phases are consistent with decahydride stoichiometry, within experimental uncertainty, and support the assignment of the composition as $(\text{La}_{0.9}\text{Y}_{0.1})\text{H}_{10}$. This conclusion is further supported by the pressure–volume trends shown in Figure S4, where the data

for both FCC and HCP phases closely follow the reported equation of state of LaH₁₀, consistent with ~10 H per metal atom.”

Comment #3:

Provide the Raman data demonstrating surplus hydrogen remained after LH.

Response:

We did not observe any hydrogen vibron after laser heating synthesis. This outcome is consistent with complete consumption of the hydrogen released from ammonia borane. Multiple cycles of pulsed laser heating were performed, which promote full uptake of available hydrogen by the alloy. The presence of a large fraction of the hexagonal P6₃/mmc phase in the recovered sample further suggests hydrogen deficiency relative to the cubic phase, again consistent with full reaction.

This interpretation is supported by other datasets

- Post-LH XRD shows only the clathrate phases (Fm-3m and P6₃/mmc), with no reflections attributable to solid H₂.
- Volume-based stoichiometry (Supplementary Note 1, Fig. S3) yields ~10 H per metal atom for both phases, matching decahydride stoichiometry and indicating full hydrogen uptake.
- Similar absence of H₂ vibrons following synthesis has been reported in prior La–H and La–Y–H studies^{4,5}, where it was attributed to complete incorporation of hydrogen into the clathrate framework.

Changes in the manuscript:

• **From Methods – Sample Preparation and High-Pressure Synthesis**

“Laser heating in both cases was performed at the interface between the metal and ammonia borane using a modulated Yb fiber laser, delivering ~300 ms focused pulses, reaching estimated temperatures of 1200–1800 K. Each synthesis involved multiple laser heating cycles, which ensured complete hydrogen uptake and phase formation.”

Comment #4:

Clarify pressure gradients across the chamber (e.g., diamond edge Raman at multiple azimuths) and map how they correlate with domain distribution.

Response:

We quantified the pressure distribution across the chamber and correlated it with the XDI maps. Figure S9 shows local pressures at the electrode–sample junctions determined from Pt(111) reflections using the Pt EOS. The electrodes span 164–170 GPa ($\Delta P \approx 6$ GPa). FCC/HCP domain boundaries do not coincide with the local pressure gradient, indicating that pressure is not the primary control on domain arrangement at synthesis conditions.

To place these results in context, we also added the center diamond-edge Raman measurement (153 GPa), which differs from the Pt-EOS values near the electrodes. Such offsets are common at ≥ 150 GPa due to differences among pressure scales and nonhydrostatic stress in XRD measurements. Accordingly, we emphasize relative ΔP rather than absolute pressure.

We note that, in this experiment, our priority was transport measurements. Raman mapping at multiple azimuths was not performed to minimize laser exposure of the diamond culets, which can trigger damage at these pressures.

Changes in the manuscript:

- **From Results – Spatial Mapping of Structural Domains via SXDM and XDI**

“The Fm-3m phase is localized in discrete clusters near Pt leads #2, #3, and #4, with the largest fraction around lead #2 where excess ammonia borane was present. By correlating optical images before and after laser heating (Fig. S3) with the XDI maps, the interface between the sample and ammonia borane is identified between leads #2 and #3, consistent with regions of higher temperature and greater hydrogen availability favoring the Fm-3m phase, while regions farther from this interface exhibit a higher fraction of the P63/mmc phase, forming a continuous matrix between leads #1 and #4. These correlations indicate that the observed structural inhomogeneity most likely arises from variations in hydrogen content and laser heating geometry. As shown in Fig. S9, the local pressure gradient does not correlate with FCC/HCP boundaries, indicating that pressure is not the dominant factor in domain arrangement at synthesis conditions.”

- **From Figure S9 – XDI maps with local pressure variation at 153 GPa**

Figure S9: XDI maps with local pressure variation at 153 GPa. A $30\ \mu\text{m} \times 30\ \mu\text{m}$ raster shows Pt leads (dark gray), cubic $Fm\text{-}3m$ (red), and hexagonal $P6_3/mmc$ (blue). Local pressures at the electrode–sample junctions were derived from Pt(111) using the Pt EOS ($K_0 = 266\ \text{GPa}$, $K_0' = 5.81$, $V_0 = 60.3793\ \text{\AA}^3$ per fcc cell). The electrodes span 164–170 GPa ($\Delta P \approx 6\ \text{GPa}$). ΔP values relative to the median (=166 GPa) are displayed on both the Pt map and the FCC+HCP overlay. Domain boundaries do not align with pressure gradients: FCC and HCP phases coexist across the full ΔP range, indicating pressure is not the primary control on domain arrangement at synthesis conditions. The center diamond-edge Raman measured 153 GPa, slightly lower than Pt-EOS values; such offsets are common above 150 GPa due to scale differences and nonhydrostatic stress. For correlation, we emphasize relative ΔP rather than absolute pressure.

References:

1. Chen, W. *et al.* Superconductivity and equation of state of lanthanum at megabar pressures. *Phys. Rev. B* **102**, 134510 (2020).
2. Storm, C. V., Roy, C. R., Munro, K. A. & McMahon, M. I. Crystal structures of lanthanum to 230 GPa. *Phys. Rev. B* **110**, 024107 (2024).
3. van der Pauw, L. J. A method of measuring the resistivity and Hall coefficient on lamellae of arbitrary shape. *Philips technical review* **20**, 220–224 (1958).
4. Somayazulu, M. *et al.* Evidence for Superconductivity above 260 K in Lanthanum Superhydride at Megabar Pressures. *Phys. Rev. Lett.* **122**, 027001 (2019).
5. Semenok, D. V. *et al.* Superconductivity at 253 K in lanthanum–yttrium ternary hydrides. *Mater. Today* **48**, 18–28 (2021).
6. Loubeyre, P. *et al.* X-ray diffraction and equation of state of hydrogen at megabar pressures. *Nature* **383**, 702–704 (1996).

X-ray Diffraction and Electrical Transport Imaging of Superconducting Superhydride (La,Y)H₁₀

A. H. Manayil Marathamkottil^{1*}, Kui Wang², Nilesh P. Salke², Muhtar Ahart², Alexander C. Mark², Ross Hrubciak³, Stella Chariton⁴, Dean Smith³, Vitali B. Prakapenka⁴, Maddury Somayazulu³, Nenad Velisavljevic^{3,5}, Russell J. Hemley^{1,2,6*}

Authors' Response to Reviewer Comments

We sincerely thank both reviewers for their time, effort, and positive evaluations of our manuscript.

Reviewer #1:

I can accept the publication of the article and have no further suggestions

Response to Reviewer #1:

We appreciate your recommendation for publication and your supportive comments.

Reviewer #2:

The authors have sufficiently addressed my concerns. I have no additional comments and recommend this manuscript for publication in Nature Communications.

Response to Reviewer #2:

We thank you for your thorough review and for confirming that our revisions have fully addressed your previous concerns. We are grateful for your recommendation for publication in Nature Communications.

This paper reports the synthesis and characterization of $(\text{LaY})\text{H}_{10}$ superhydrides. Two coexisting cubic $Fm\text{-}3m$ and hexagonal $P63/mmc$ phases were observed over the studied pressure range. The μm -scale structural inhomogeneity was revealed and visualized via SXDM and XDI. Four-probe DC resistance measurements confirmed superconductivity with two distinct transitions. The first T_c is 244 K associated with the cubic phase and the second T_c near 220 K linked to the hexagonal phase. However, a few points need clarification before it can be recommended for acceptance.

1. Why the number and shape of the XRD diffraction peaks are different at 158 GPa and 172 GPa for $\text{La}_{0.9}\text{Y}_{0.1}$? such as the peaks at 7, 10, and 13.5 degrees at 158 GPa.

2. Although the author used the Le Bail method for fitting, several peaks remain unknown, as shown in the figure below. Moreover, the fitting deviation for the 102 peak is excessively large. Could these peaks originate from a new structural phase? Additionally, the authors should provide the XRD diffraction pattern alongside the integrated intensity plot.

3 I suggest changing "DAC#2" to "DAC#1" in the main text to enhance clarity for readers.

4 Why do the positions of the unknown peaks differ between 153 GPa and 136 GPa in DAC#2? What is the origin of these unknown peaks? Were the XRD patterns collected from the same sample location for these two measurements? The XRD diffraction pattern and micrograph of the collection positions should be provided.

5, In the main text, the author named the sample as (La0.9,Y0.1)H10, How did the author arrive at this ratio? And What evidence supports the successful incorporation of Y for the reaction product?

7, The images in Figure 2 at 153 and 136 GPa should be presented in the same orientation to facilitate direct comparison and interpretation.

8, The Raman spectroscopy data used for pressure calibration should be

provided in the supplementary materials.